# Prediction of Anthocyanin Color Stability against Iron Co-Pigmentation by Surface-Enhanced Raman Spectroscopy

**DOI:** 10.3390/foods11213436

**Published:** 2022-10-29

**Authors:** Haochen Dai, Adam Forbes, Xin Guo, Lili He

**Affiliations:** 1Department of Food Science, University of Massachusetts Amherst, Amherst, MA 01003, USA; 2Department of Chemistry, University of Massachusetts Amherst, Amherst, MA 01003, USA

**Keywords:** anthocyanin, iron, co-pigmentation, color, stability, Raman, SERS

## Abstract

The color change resulting from anthocyanin and iron co-pigmentation has been a significant challenge for the food industry in the development of many iron-fortified foods. This present study aims to establish a quantitative model to predict the degree of color stability in the presence of dissolved iron using surface-enhanced Raman spectroscopic (SERS) spectra. The SERS spectra of anthocyanin extracts from seven different plant sources were measured and analyzed by principal component analysis (PCA). Discrimination among different sources of anthocyanin was observed in the PCA plot. Different stability indexes, obtained by measuring both the color intensity stability and color hue stability of each sample, were established based on UV–vis analysis of anthocyanin at pH 3 and 6 with and without ferric sulfate. Partial least square (PLS) regression models were applied to establish the correlation between SERS spectra and stability indexes. The best PLS model was built based on the stability index calculated from the bathochromic shift (UV–vis spectral range: 380–750 nm) in pH3 buffer and the SERS spectra, achieving a root mean square error of prediction (RMSEP) of 2.16 nm and a correlation coefficient value (R^2^) of 0.98. In conclusion, the present study developed a feasible approach to predict the stability of anthocyanin colorants against iron co-pigmentation. The developed method and models can be used for fast screenings of raw ingredients in iron-fortified food products.

## 1. Introduction

The color of food products is one of the most important sensorial attributes for consumers’ preferences. Despite artificial colorants being more stable and cheaper than natural alternatives, the trend of replacing artificial colorants with natural ones is demanded by consumers. Studies of artificial colorants demonstrated their ability to induce allergic reactions in human children or potential carcinogenic effects when consumed in large quantities in some animal studies [1,2,3,4,5].

Anthocyanins are the most abundant natural colorants and are found in thousands of plants, in addition to carotenoids, chlorophylls, and betalains. The anthocyanin compounds are chemically defined as glycosylate polyhydroxy or poly methoxy derivatives of 2-phenylbenzopyrilium with two benzyl rings, as shown in Figure 1 [6,7,8]. The major anthocyanins and their plant sources can be found in Table 1 [8,9,10,11,12,13,14].

The color stabilities of extracted anthocyanins are affected by many factors, such as pH, heat, light, the presence of other chemical compounds, and metal ions. Given that many food manufacturers are fortifying their products with iron to fight global iron deficiency, especially for baby food, metallic ions are considered one of the leading compounds affecting the color stability of isolated anthocyanins within the food matrix [15,16,17,18,19,20,21,22]. The major reaction involved between anthocyanin and metallic ions is called co-pigmentation. The anthocyanin–metal complex is formed due to the metal-binding capability of the ortho-dihydroxyl groups on the B ring (Figure 1) [8]. Most anthocyanin–metal co-pigmentation products are always having a color shift to blue or violet hues [23]. However, this reaction product is not favorable in most food matrices, where anthocyanins are selected and preferred as red colorants. Thus, the organoleptic changes caused by the fortification of iron should be seriously investigated.

In the past few decades, the interaction between anthocyanin and ferrous/ferric ions has been studied extensively. In addition, a considerable amount of literature has been published on the mitigation of metal-induced color shifts of anthocyanin. Data from several groups suggest that acylated anthocyanins are consistently more stable against pH change and iron fortification [24,25,26,27]. It is believed that the acylated group protects the chromophore from nucleophilic attack or the attack from ferrous/ferric ions [27].

The conventional UV–vis method is one of the most adopted methods for anthocyanin–metal studies [28,29,30]. The chromatic changes of anthocyanin solutions after metal co-pigmentation can be recognized as shifts in maximum absorption bands or shifts in absorbance values. Despite the relative simplicity and great popularity of the UV–vis method, some limitations of this method still hinder researchers from a deeper understanding of the interaction between anthocyanins and metal ions. For instance, the method requires tedious preparation work to quantify and standardize the concentration of all anthocyanin samples before iron salts are incorporated into the solution. Second, this method lacks good specificity to identify or characterize complex anthocyanin mixtures from different plant origins. Thirdly, UV–vis spectra failed to provide useful molecular structure information, which is critical to anthocyanin color stability against iron co-pigmentation.

The objective of this study is to develop an approach that can analyze anthocyanin extracts from different plant sources and establish a prediction model for the color stability of anthocyanins with the incorporation of iron by means of surface-enhanced Raman spectroscopy (SERS). SERS, a technique based on Raman spectroscopy utilizing nanoscale metallic surfaces to enhance the Raman scattering, has shown good capabilities in analyzing the molecular fingerprints of anthocyanin samples compared to normal Raman spectroscopy in terms of higher characteristic peaks and less fluorescence interference [31].

The hypothesis is there are underlying correlations between the SERS spectral fingerprints and the metal-induced color shifts of various anthocyanins. In the present work, co-pigmentations between anthocyanin extracts (*n* = 9) and iron salt were evaluated, and their color stabilities were translated into different stabilization indexes. Ferric ion was chosen due to better stability and high reactivity with anthocyanin to achieve bathochromic shifts at low concentrations [28,29,32]. High concentrations of metal ions may result in the formation of complex precipitation, which is not ideal for UV–vis spectroscopy analysis [32]. SERS spectra of purified anthocyanin extracts were collected, and a mathematical model was built for the prediction of anthocyanin color stability based on the SERS data. Moreover, diagnostics of peak assignments were carried out to explain the relationship between SERS spectra and the color stability of anthocyanin extracts.

To our best knowledge, this approach was not achieved elsewhere. The outcome of this study has great potential to be used in real applications in raw material screenings for product development or quality control.

## 2. Materials and Methods

### 2.1. Raw Material and Chemical Reagents

Plants extracts (7 different plant sources, *n* = 9) were provided by three suppliers: Exberry (Dallas, NC, USA), Food Ingredient Solutions (Teterboro, NJ, USA), and Charles Bowman Co. (Holland, MI, USA). Received powdered or liquid samples were kept in the dark at 4 °C until analysis. All samples were tested within 1 month after the reception. The choice of plants was made to maximize the structural differences of the dominant anthocyanin, which are based on their degree of acylation and their B-ring structures.

All chemical reagents are analytical grade unless specifically mentioned. Silver nitrate (AgNO_3_) (assay 99.9%), potassium chloride (assay 99.8%), methanol (assay 99.8%) (Ward Hill, MA, USA), hexanes (assay 99.9%), acetonitrile (assay 99.9%), hydrochloric acid (min. 37%) were purchased from ThermoFisher (Fair Haven, NJ, USA); polyethylenimine (branched, average mv ~25,000), ethyl acetate (assay 99.8%), iron (iii) sulfate hydrate (assay 97%), sodium acetate (assay ≥ 99%) from Sigma-Aldrich (St. Louis, MO, USA). C-18 solid-phase extraction (SPE) columns (500 mg sorbent) were obtained from ThermoFisher Scientific (Rockland, TN, USA). All buffers and solutions were prepared using ultrapure water (Micropure, TheroFisher Scientific, Langenselbold, Hungary). All buffers used in this study were monitored with an accumet AE 150 pH meter (ThermoFisher Scientific).

### 2.2. Sample Preparation

A modified anthocyanin purification protocol was adopted [28,33]. Each sample was dissolved or diluted in acidified water (1% *w*/*w*) before being filtered through a PES filter (0.2 µm, Whatman, Buckinghamshire, U.K.). The extract was then loaded in a C-18 SPE column, pre-conditioned with 3 mL acidified methanol (0.01% HCl) and 3 mL anthocyanin and acidified water (0.01% HCl) solution. First, the column was washed with 9 mL of acidified water (0.01% HCl), followed by 9 mL of ethyl acetate to remove organic acids, free sugars, and phenolic compounds, which can work as co-pigments with iron. Next, the anthocyanin fraction was eluted with 3 mL acidified methanol (0.01% HCl). The collected eluent was concentrated by a HeizbadHei-UAP rotary evaporator (Heidolph Instrument, Schwabach, Germany) at 40 °C. Finally, concentrated anthocyanin stock solutions (acidified MeOH) were flushed with gentle Helium streams and then stored under −20 °C until further analysis.

### 2.3. Anthocyanin Quantitation (pH Differential Method)

All anthocyanin stock solutions were quantified using the pH differential method [34]. In brief, each stock anthocyanin solution was first diluted in 0.025 M potassium chloride buffer (pH 1) and 0.4 M sodium acetate buffer (pH 4.5). Then, each buffer solution was measured in a SpectraMax M2 spectrometer (Molecular Devices, Sunnyvale, CA, USA) under 520 nm and 700 nm, respectively. Based on the absorbance difference, the concentration (cyanidin-3-glucose equivalent, C3G) of each stock solution can be calculated using the equation below,
$$C_{\left(\mathrm{mg}/L\right)}=\frac{A\times MW\times DF\times 1000}{\varepsilon \times 1}$$
where *A* equals (A520-A700) pH_1.0_ − (A520-A700) pH_4.5_, *MW* is the molecular weight for C3G (449.2 g/mol), *DF* is the total dilution factor, 1 stand for the curvet pathlength (1 cm), *ε* is the molar extinction coefficient in for C3G (26,900 L × mol^−1^ × cm^−1^), 1000 is the factor conversion from g to mg.

### 2.4. Iron Stabilization Index Acquisition

Stock anthocyanin solutions were then diluted to 30 µM in 0.1 M pH 3 and 6 sodium acetate buffers. Iron (iii) sulfate was diluted in ultra-pure water at 200 mM and then diluted to 0.6 mM in both buffers. The iron solutions were then added to the anthocyanin solution with a 1:10 (ACN: Fe) ratio under their corresponding pH condition. The control sample was analyzed with the addition of buffers instead of iron solutions. The pH level of all tested samples was carefully monitored. Samples were mixed well and kept in the dark prior to the analysis of visible transmittance (SpectraMax M2, 380–700 nm, step size 5 nm). Duplicates were evaluated for each sample.

The stabilization index was assigned to the bathochromic and hyperchromic shifts at both pH levels. As shown in Figure 2, index 1 and 2 corresponds to the bathochromic shift at pH 3 and 6. Similarly, index 3 and 4 translates to the hyperchromic shifts with the incorporation of iron at both pH conditions. Indexes 5 and 6 are calculated below as the root mean square of the conjunct bathochromic shifts and hyperchromic shifts.
$$Index\;5=\sqrt{\frac{\left(Index\;1^{2}+Index\;2{\;}^{2}\right)}{2}}\;;\;Index\;6=\sqrt{\frac{\left(Index\;3^{2}+Index\;4{\;}^{2}\right)}{2}}$$

### 2.5. Silver Colloid Synthesis

A silver colloid solution was prepared using branched polyethylenimine as a reducer with slight modification [35]. Briefly, 200 mL of 0.2% silver nitrate and 200 mL of 0.2% branched polyethylenimine, were gently mixed with a 1:1 volume ratio. Then, the mixture was exposed under a UV lamp (365 nm, Analytik Jena, Upland, CA, USA) in dark for 8 h with continuous stirring at 500 RPM on an Isotemp stirring hotplate (Fisher Scientific Co. LLC, Waltham, MA, USA). The residual polymers were then removed from the silver colloid solution by washing the nanoparticle three times. To complete this process, the developed solution was transferred equally in two 250 mL centrifuge tubes and centrifuged at 17,000× *g* for 5 min (Sorvall Lynx 400 centrifuge, ThermoFisher Scientific, Amkalkberg, Germany). The supernatant was removed, and ultra-pure water was added to reconstitute the volume and repeat the centrifugation process. The final silver colloid solution was concentrated to 200 mL (32.5 mg/L) by removing the surplus supernatant after the third centrifugation and stored at 4 °C in dark for further use. Detailed characterization of the fabricated silver nanoparticles was shown in Appendix A. In brief, the UV–vis spectrum (Appendix A) of the fabricated nanoparticles displayed the characteristic surface plasmon resonance (SPR) band for silver nanoparticles, which was centered at 410 nm. Dynamic light scattering analysis and scanning electron microscopic analyses were conducted for hydrodynamic diameter (67.66 ± 4.27 nm, Appendix A) and geometric diameter (54.5 ± 18.2 nm, Appendix A), respectively. Lastly, the SERS capability of the fabricated silver nanoparticles was compared against commercial 40 nm silver nanospheres (NanoComposix Inc., San Diego, CA, USA) and citrate-reduced nanospheres [36]. The result (Appendix A) demonstrated great SERS performance in using the fabricated silver nanospheres for anthocyanin characterization.

### 2.6. Anthocyanin SERS Reference Spectra Acquisition

Concentrated silver colloid solution (10×) was prepared by removing the supernatants after centrifugation (10,000 rpm 5 min, ThermoFisher Scientific). Then, acidified water (0.01% HCL) was used to reconstitute the concentrated silver colloid solution to 1× to prevent pH fluctuations in the anthocyanin solution after adding the silver colloid solution. Each anthocyanin stock solution was diluted to 56 mg/L with acidified water (0.01% HCl) to obtain good SERS signal with reduced fluorescent interference from the sample. For SERS analysis, 100 µL of the diluted anthocyanin solution was fully mixed in a test tube with an equal volume of acidified silver colloid solution. A single droplet of the mixture (0.5 µL) was dropped on an aluminum foil-coated glass slide in dark and air-dried prior to analysis. To ensure good reproducibility, anthocyanin SERS spectra were collected from at least two independent preparations. In each preparation, at least three droplets were analyzed. Representative 20 spectra for each anthocyanin sample were collected using a DXRi Raman imaging microscope (ThermoFisher Scientific, Madison, WI, USA) equipped with a 780 nm laser. Each measurement was performed under a laser power of 5 mW and an integration time of 1 s in a spectral range of 300–2000 cm^−1^.

### 2.7. Data Analysis

Spectral data from the UV–vis spectrometer and the Raman imaging microscope were processed in the OMNIC software (version 9.7.46, ThermoFisher Scientific). Principal components analysis (PCA) and partial least square regression (PLSR) were executed in the TQ Analyst software (version 9.7.0.27, ThermoFisher Scientific). Standard normal variate (SNV) path length (spectral range 400–1800 cm^−1^) was used for correction after applying second derivatives for the anthocyanin SERS spectra for both PCA and PLSR analysis. Spectral regions 370–918 cm^−1^ and 1099–1692 cm^−1^ were picked for PCA and PLSR analysis based on the distribution of characteristic peaks of anthocyanin. In PCA, the calibration set consisted of a total of 180 spectra, with 20 spectra of each class. In PLSR, the X data matrix included the SERS spectra within the selected spectral range, and Y consisted of the stabilization index for the corresponding SERS spectra. The result was expressed as prediction vs. actual in the PLSR plots. The performance of the PLSR model is evaluated by the root mean square error of cross-validation (RMSECV, leave-one-out cross-validation), root mean square error of prediction (RMSEP) values, and their corresponding correlation coefficient values (R^2^), where the optimization of latent variables (factors) was also confirmed using the prediction residual error sum of square (PRESS) function for each stabilization indexes. For the calculation of RMSEP for internal validations, 180 spectra were divided into training groups (75%, N = 135) and validation groups (25%, N = 45) for each index. The optimal model was chosen with the highest values of R^2^ and the minimum of RMSECV/RMSEP.

## 3. Results and Discussion

### 3.1. Stabilization Index Analysis of Anthocyanin Extracts against Iron Incorporation

The stabilization indexes of each anthocyanin are summarized in Table 2. The larger the absolute value of each index indicates greater hue or intensity shifts with the addition of iron. In other words, lower color stabilities of anthocyanins are associated with increasing index values. Generally, greater index values occur at pH 6 (index 2 and 4) rather than pH 3 (index 1 and 3). This phenomenon was due to an increased number of iron chelations as well as anthocyanin self-associations with increasing pH and was well observed in other studies [8,26]. Comparing bathochromic (Index 1 and 2) and hyperchromic (Index 3 and 4) shifts, a strong correlation was detected among these indexes. Samples with low bathochromic shifts usually exhibited low hyperchromic shifts. These data indicated that anthocyanin extracts from different plant sources exhibit different stability indexes against iron incorporation. With stable structure, such as degree of acylation or B ring structure, anthocyanin molecules are less likely to form anthocyanin–metal complexes or self-association.

Index 5 and 6 (root mean square of indexes under pH 3 and 6) represent the overall stability under two different pH conditions. It was observed that acylated anthocyanins (purple sweet potato, purple carrot, red radish) have higher scores 1.29–33.97 for index 5 and 0.112–0.249 for index 6, respectively than those that are known to have less or little acylation (grape juice, elderberry, acai, black currant), ranging from 16.96–36.53 for index 5 and 0.199–0.401 for index 6 [33,37,38,39,40,41]. The result from this study aligns with the studies in the literature, where acylated anthocyanins tend to show better resistance than non-acylated anthocyanins against co-pigmentation from ferric ions due to the protected chromophore in acylated anthocyanins [24,25,26,27].

According to index 5, which can translate to color hue difference, the overall color stability against iron from high to low are as follows: red radish > purple sweet potato > grape juice color > purple carrot > elderberry, acai, and black currant. Grape juice is generally considered to consist of non-acylated anthocyanins, such as Malvidin-3-O-glucoside or Peonidin-3-O-glucoside [39,41]. However, it should be noted that both of these anthocyanins only contain one hydroxy group on the B ring, which is not possible for metal chelation [29]. Further evidence also indicates that the composition of some grape juices may vary significantly. For instance, acylated anthocyanin, Cyd 3-(p-coumaroyl) glucoside 5-glucoside, was found to be the dominant anthocyanin in Campbell Early (Vitis labrusca) grape juice [39]. Thus, considering its fair color stability in Table 2, it is reasonable to speculate that the grape juice color tested in the present study may contain acylated anthocyanin as the dominant anthocyanin. However, the detailed composition analysis of each extract is not the main objective of this study.

It is worth mentioning that each purified extract in the present study is a mixture of anthocyanin varieties whose chemical composition may be drastically different due to different glycosylation or acylation in different plant extracts. However, for anthocyanin extracts from the same plant sources, its primary anthocyanin contents were reported to be very similar unless great varieties of plant species were involved [13,14,28,33,37,38,39,40]. Thus, it is not surprising to see that anthocyanin extracts from the same plant sources (purple sweet potato and purple carrot) showed very similar bathochromic shifts (index 1, 2, and 5). This indicates that the composition of the extracts from the same plant source is relatively similar using the applied purification protocol, and the index of the mixture could be used to represent the overall anthocyanin stability of certain plants. Studying the extracts rather than individual anthocyanin is beneficial for food companies to evaluate the quality of the colorant extracts from their suppliers.

### 3.2. SERS Characteristics of Anthocyanin Extracts

The SERS spectra of purified anthocyanin extracts are summarized in Figure 3, with their corresponding Raman shift listed in Table 3. Although the tested anthocyanin extracts have different chemical compositions, they share some typical characteristic peaks. Such patterns can be used to identify anthocyanin. Specifically, the peaks at ~1330, 1370, 1530, 1570, 1600, and 1630 cm^−1^ are associated with ring stretching vibration modes of the A and B rings [42,43,44]. The intense peaks at ~1600 cm^−1^ are related explicitly to the flavylium cationic form of the anthocyanins [44]. In addition, other strong peaks at ~1330 cm^−1^ are attributed to the inter-ring bond stretching modes [42,43,44]. The wavenumber of this band is closely related to the π-electron interaction between the B ring and the rest of the molecule. For instance, a similar study has reported that the wavenumber is positively correlated with the increase in the delocalization of the π-electron [43]. Two narrow bands at ~1080 and 1240 cm^−1^ are associated with the stretching of the C-O bond, whereas the bands at ~1190 cm^−1^ are related to the bending modes of the hydroxyl groups. In addition, some skeleton in-plane and out-of-plane bending modes can be recognized at ~420, 480, 630, 640 cm^−1^.

### 3.3. PCA Analysis of Anthocyanin SERS Spectra

The 3-D principal component scores (PCA) of all tested anthocyanins can be found in Figure 4. The PCA result is a reflection of the spectral differences from 370–1670 cm^−1^. In addition, SNV and second derivative filters were applied to normalize the spectra, separate overlapping peaks, and reduce baseline shifts. Anthocyanins that generate similar SERS patterns from Figure 3 tend to form closely positioned or overlapping clusters within the coordinate system. It is obvious from Figure 4 that sample spectra from the same ingredient formed overlapping clusters within each other. For instance, purple sweet potato from supplier 1 (S1) had overlapping clusters with purple sweet potato from supplier 2 (S2). The same phenomenon applied to purple carrot samples from suppliers 1 and 2 (S1 and S2), respectively.

The result also validated the conclusion in Section 3.1 that anthocyanin extracts from the same plant source exhibited similar stabilization indexes. In this case, anthocyanin extracts derived from the same plant exhibited similar SERS signals. This finding is well expected since anthocyanin extracts from the same plant source tend to be chemically similar in the dominant anthocyanins and anthocyanin compositions [45]. Such similarities will unsurprisingly generate similar SERS patterns.

Along the PC 1 direction, it is clear that the data points from the stable anthocyanins (red radish, purple sweet potato, and grape juice) were separated from less stable ones (purple carrot, elderberry, acai, and black currant), which is highly in accordance with the result in Table 2. This trend indicates a correlation between the SERS spectra of anthocyanin extracts and their corresponding color stability.

To further analyze the characteristic peaks that contribute to the variation, the principal component spectra were presented in Appendix A. Six contributing SERS bands were spotted at ~1610, 1330, 730, 715, 637, and 548 cm^−1^, which correspond to the vibration modes in the A ring, B ring, and skeleton bending modes of anthocyanin.

### 3.4. PLSR Analysis between SERS Spectra and Stabilization Index of Each Anthocyanin Extract

The descriptive statistics of the PLS model for each index are summarized in Table 4. Additionally, the correlation between the collected and predicted stabilization index values from external validations can be seen in Appendix A. From the result in Table 4, it can be concluded that the SERS spectra showed a satisfactory R^2^ in the cross-validation for index 1 (0.91), index 2 (0.86), and index 5 (0.91). In addition, the RMSECV values were reported to be 4.60, 9.26, and 5.94 nm for indexes 1, 2, and 5, respectively. On the other hand, poor performances were observed in cross-validation models with indexes 3, 4, and 6, only achieving R^2^ values of 0.51, 0.17, and 0.13, respectively. A similar trend was observed in the external validation, where indexes representing bathochromic shifts (index 1, 2, 5) had higher R^2^ than those translating to hyperchromic shifts (index 3, 4, 6). Therefore, it is suggested that the models of bathochromic performed better than those for hyperchromic shifts, possibly because the color intensity is relatively less stable than the color hue shift during the UV–vis measurements.

The results of the internal validation model in Table 4 are reflected in Appendix A. In detail, index 1 achieves the best fitting to the prediction vs. the actual line, whereas index 4 generates the worst fitting plots. After comparing all PLS plots, a similar conclusion can be made: bathochromic indexes (index 1, 2, and 5) have better overall fittings than those of hyperchromic indexes (index 3, 4, and 6). The result in index 1 further confirms our initial hypothesis that there is a close correlation between anthocyanins’ SERS spectra and their color stability with iron incorporation. One can potentially use the PLSR model to evaluate the stability indexes of an unknown anthocyanin sample by analyzing its SERS spectrum.

### 3.5. Study of Anthocyanin’s Iron Stability on SERS Signal

According to Table 4 and Appendix A, index 1 is chosen to be the best model for anthocyanin color stability prediction. Hence, in Figure 5, the PLS loading spectra of index 1 are studied to reveal the contributing Raman bands for the color stability prediction of anthocyanin. In the second derivative spectra, peaks pointing downwards suggest a positive influence on the development of the equation.

The major bands to predict anthocyanin stability are mainly related to anthocyanins’ structural vibration modes. From the result in factors 1 and 2, the peak at ~1300 and 1330 cm^−1^ are closely related to the quantification and prediction of anthocyanin’s color stability. Given that the ring’s stretching modes are closely responsible for the signal within the spectral range of 1099–1692 cm^−1^ [43], the chromophore status can be reflected based on the SERS signal within this range. It is worth remembering that the 1330 cm^−1^ band is the inter-ring stretching mode, whose wavenumber and peak intensity are greatly affected by the location, type, and the number of substituents. A shift to higher wavenumbers was observed in most stable anthocyanin samples, including red radish (1339 cm^−1^), purple sweet potato (1335, 1336 cm^−1^), and grape juice color (1343 cm^−1^). On the other hand, the wavenumber of this peak in most unstable anthocyanins was relatively smaller at ~1329 cm^−1^. As previously discussed, this increase in wavenumber may be related to the delocalization of the π-electron caused by the substituents in the molecule [43]. Thus, it is reasonable to suggest that anthocyanin with an elevated wavelength at ~1330 cm^−1^ tends to exhibit higher color stability against iron, possibly due to a larger and more complex molecular structure.

In addition, the band at ~1300 cm^−1^ may be associated with the vibration modes among the glucoside and acylated groups within anthocyanin molecules. The presence of this peak in the SERS spectrum is highly in accordance with good color stability, especially for red radish extracts, which are known to obtain a high amount of acylated anthocyanins. SERS spectra (Figure 3) of other samples with acylated anthocyanin, i.e., purple sweet potato, purple carrot, and grape juice color, also confirm this suggestion. The only exception is the purple carrot extract, whose dominant anthocyanin is believed to be cyanidin-3-O-(2″-xylose-6″-glucose-galactoside) by acylation with sinapic, ferulic, and p coumaric acid [13,46]. One possible explanation is that most purple carrot anthocyanins contain only one glycosylation position in the 3-position on the C ring. In contrast, other anthocyanins (red radish, purple sweet potato, grape juice) are believed to have sugar moieties connected in the 3 and 5 positions in the flavylium group. Furthermore, purple carrot does not contain a high concentration of acylated anthocyanins (6.4 mg/g), which is significantly lower than that of red radish (116.1 mg/g) [25].

Peaks around 720, 640, and 540 cm^−1^ are also strongly related to the prediction of color stability according to the PLS loading spectrum (factor 1). These three bands correspond to the vibration modes of the aromatic system, C-C skeleton in plant bending and C-C in-plane bending in anthocyanin molecules, respectively [42,43,44]. Despite no apparent trends that can be concluded from the anthocyanin SERS spectra in Figure 3, most of these peaks closely relate to anthocyanin’s skeleton structural vibration modes. At last, the peaks at ~1610, 1520, and 1240 cm^−1^ showed a positive correlation as well. However, the significance of these bands is relatively lower than those already discussed bands.

## 4. Conclusions

The present work demonstrated a viable approach to both distinguish anthocyanin sources and predict the iron-induced color stability of extracted anthocyanins with sufficient accuracy by means of SERS coupled with multivariant analysis. In addition, a diagnostic of SERS signals was executed that allowed the identification of color-stable anthocyanin with the presence of iron. Based on the tested result, the anthocyanin extracts from different plant sources can be easily differentiated using collected SERS spectra coherently with their stability index. Next, a good PLS model was built with an R^2^ of 0.98 and an RMSEP of 2.16 nm to predict color stability using external validations. At last, the bands (~1330 cm^−1^) assigned to the inter-ring stretching modes and the bands reflecting skeleton vibration modes closely correlate to the anthocyanins’ color stability predictions. One possible explanation is that these bands are highly coherent with the delocalization of the π-electron and the bending modes of the overall anthocyanin skeleton.

From the result of this work, it is demonstrated that SERS can be used as a rapid and versatile tool for screening raw anthocyanin sources together with the characterization of their color stability against iron incorporation. In addition, this study has established, for the first time, that a good correlation between anthocyanin’s stability indexes and SERS signal can be achieved for prediction. It also demonstrated fundamental discussions regarding anthocyanin to iron co-pigmentation from a Raman perspective. In addition, this approach greatly reduces the analysis time. The traditional way of assessing iron-induced anthocyanin color shifts involves tedious protocols, including anthocyanin solutions preparation, anthocyanin concentration standardization, incubation with iron salt solutions, recording UV–vis spectra of target sample and control sample, and calculating the bathochromic shifts [28]. With the developed approach, one only needs to obtain the SERS spectra after preparing anthocyanin solutions to assess the color stability of anthocyanin in the presence of iron. Moreover, the outcome of this study shows great potential for real-world applications, facilitating anthocyanin suppliers’ quality control protocols and assisting food developers with their raw material verification and assessment.

## Figures and Tables

**Figure 1 foods-11-03436-f001:**
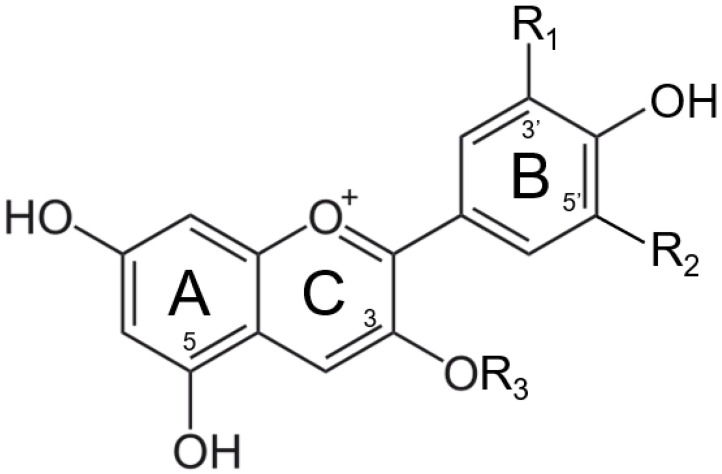
Structure of major anthocyanin-3-O-glucoside (R3).

**Figure 2 foods-11-03436-f002:**
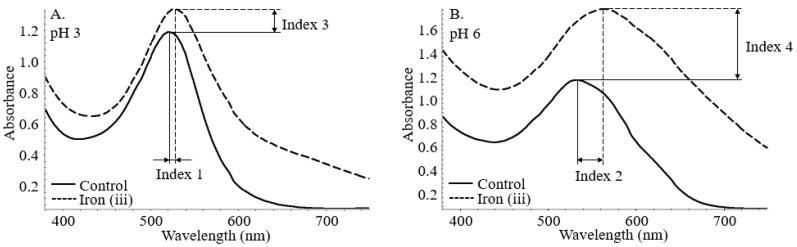
Bathochromic (Index 1 (**A**) and 2 (**B**)) and hyperchromic (Index 3 (**A**) and 4 (**B**)) shifts of co-pigmented anthocyanin measured by UV–vis spectroscopy.

**Figure 3 foods-11-03436-f003:**
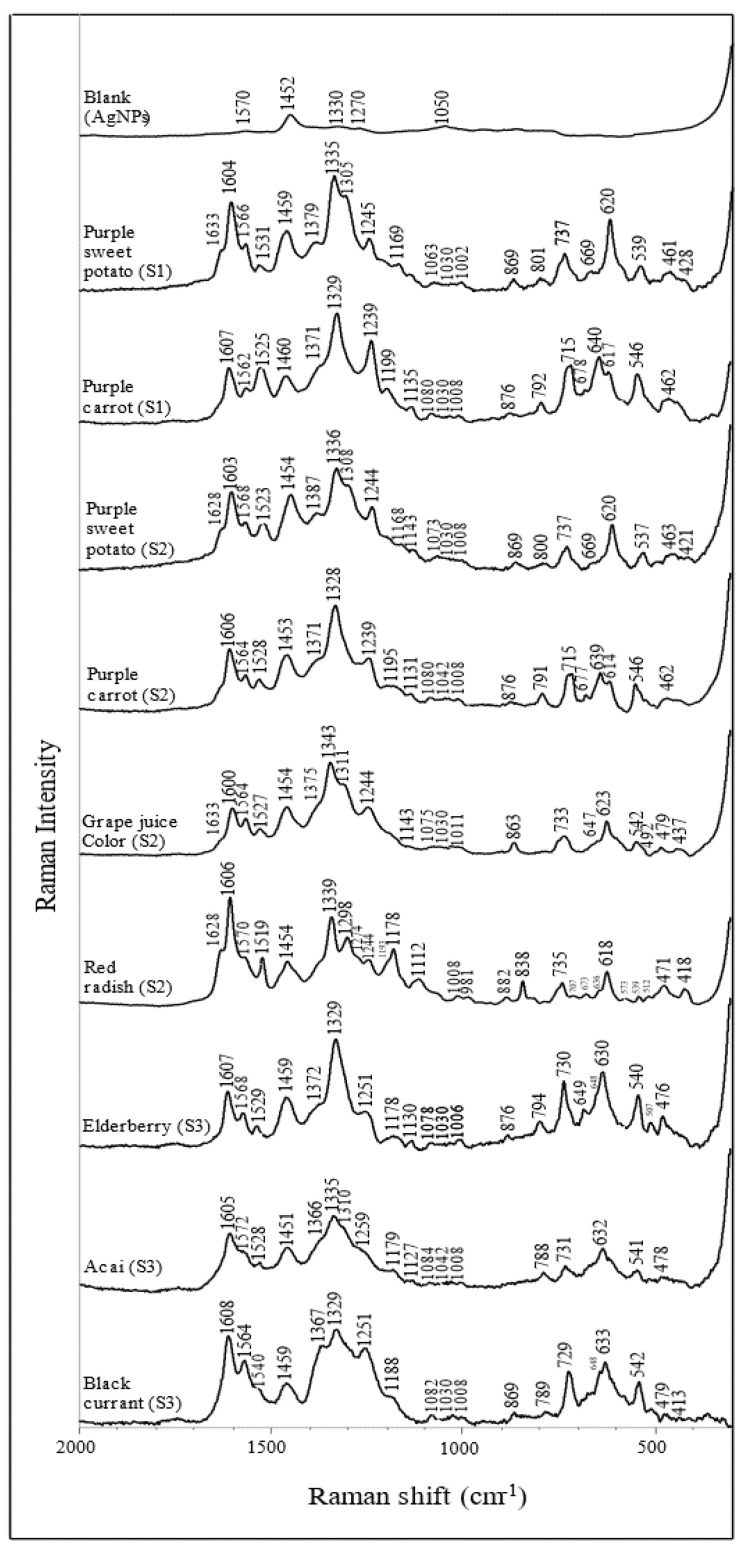
Surface−enhanced Raman spectra of anthocyanin extracts from different plant sources (dried acidified solution).

**Figure 4 foods-11-03436-f004:**
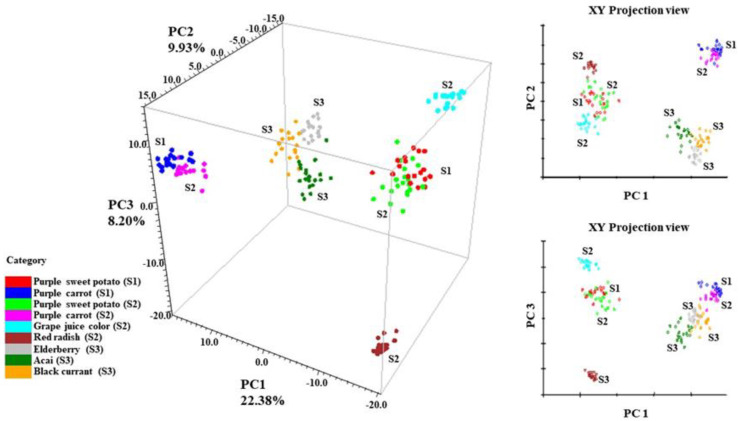
PCA scores of collected anthocyanin SERS signal, S1 (Supplier 1); S2 (Supplier 2); S3 (Supplier 3).

**Figure 5 foods-11-03436-f005:**
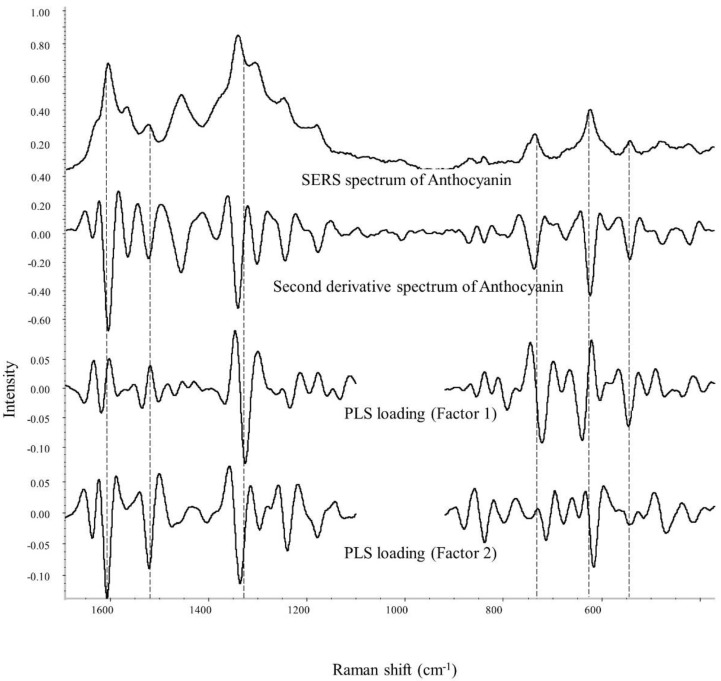
PLS loading spectra for stabilization index 1 using anthocyanin’s SERS signal.

**Table 1 foods-11-03436-t001:** Major anthocyanin structures and plant sources.

Anthocyanin	Abbrev.	R_1_	R_2_	Plant Sources	References
Cyanidin	CYN/Cy	OH	H	Acai, black currant, elderberry, purple carrot, purple sweet potato	[8,11,12,13]
Delphinidin	DEL/Dp	OH	OH	Black currant, blood orange, purple carrot, red cabbage	[8,10,11,13]
Petunidin	PET/Pt	OH	OCH_3_	Bilberry, grape	[9,11]
Pelargonidin	PEL/Pg	H	H	Acai, red radish, strawberry	[8,14]
Peonidin	PEO/Pn	OCH_3_	H	Grape, purple sweet potato	[9,11]
Malvidin	MAL/Mv	OCH_3_	OCH_3_	Bilberry, grape	[8,9,11]

**Table 2 foods-11-03436-t002:** Stabilization indexes of collected anthocyanin extracts.

Anthocyanin Extracts	Stabilization Indexes (Mean ± SD)					
	Index 1	Index 2	Index 3	Index 4	Index 5	Index 6
Purple sweet potato (S1)	2.64 ± 1.54	11.64 ± 0.06	−0.242 ± 0.054	0.196 ± 0.062	8.47 ± 0.28	0.220 ± 0.057
Purple carrot (S1)	24.59 ± 1.22	41.26 ± 0.49	0.048 ± 0.004	0.349 ± 0.031	33.97 ± 0.14	0.249 ± 0.022
Purple sweet potato (S2)	2.43 ± 1.13	7.37 ± 1.18	−0.149 ± 0.041	0.144 ± 0.003	5.50 ± 1.04	0.147 ± 0.022
Purple carrot (S2)	26.70 ± 0.06	32.75 ± 0.16	−0.0395 ± 0.013	0.236 ± 0.018	29.88 ± 0.06	0.169 ± 0.014
Grape juice color (S2)	−3.48 ± 0.13	23.73 ± 0.28	0.099 ± 0.015	0.558 ± 0.003	16.96 ± 0.21	0.401 ± 0.001
Red radish (S2)	−1.26 ± 0.25	1.31 ± 0.01	−0.111 ± 0.083	0.113 ± 0.016	1.29 ± 0.13	0.112 ± 0.031
Elderberry (S3)	22.57 ± 0.35	44.72 ± 1.23	0.089 ± 0.004	0.413 ± 0.015	35.42 ± 0.67	0.298 ± 0.010
Acai (S3)	5.35 ± 0.24	51.38 ± 2.12	0.085 ± 0.025	0.261 ± 0.181	36.53 ± 1.47	0.199 ± 0.113
Black currant (S3)	13.75 ± 1.39	47.71 ± 0.39	0.098 ± 0.006	0.450 ± 0.013	35.11 ± 0.54	0.326 ± 0.008

Note: S1 (Supplier 1); S2 (Supplier 2); S3 (Supplier 3); SD (Standard deviation), low color stabilities of anthocyanin are associated with high absolute value of each stabilization index.

**Table 3 foods-11-03436-t003:** Band wavenumbers and their tentative assignments of anthocyanin extracts.

Purple Sweet Potato (S1)	Purple Carrot (S1)	Purple Sweet Potato (S2)	Purple Carrot (S2)	Grape Juice Color (S2)	Red Radish (S2)	Elderberry (S3)	Acai (S3)	Black Currant (S3)	Blank (AgNPs)	Tentative Assignment
1633 (sh)		1628 (sh)		1633 (sh)	1628 (sh)					ν (ring A + B)
1604 (s)	1607 (s)	1603 (s)	1606 (s)	1600 (s)	1606 (s)	1607 (s)	1605 (s)	1608 (s)		ν (ring A + B)
1566 (sh)	1562 (sh)	1568 (sh)	1564 (sh)	1564 (s)	1570 (sh)	1568 (m)	1572 (sh)	1564 (m)	1570 (w)	ν (ring B)
1531 (w)	1525 (s)	1523 (m)	1528 (m)	1527 (m)	1519 (s)	1529 (m)	1528 (w)	1540 (sh)		ν (ring A + B)
1459 (s)	1460 (s)	1454 (s)	1453 (s)	1454 (s)	1454 (s)	1459 (s)	1451 (s)	1459 (s)	1452 (s)	
1379 (sh)	1371 (sh)	1387 (w)	1371 (sh)	1375 (sh)		1372 (sh)	1366 (sh)	1367 (sh)		ν (cc)
1335 (s)	1329 (s)	1336 (s)	1328 (s)	1343 (s)	1339 (s)	1329 (s)	1335 (s)	1329 (s)	1330 (w)	ν (ring B); δ (ch)
										ν inter-ring coupled
										with a vibration of
										γ-pyrone
1305 (sh)		1308 (sh)		1311 (sh)	1298 (s)		1310 (sh)			
					1274 (sh)				1270 (w)	
1245 (m)	1239 (s)	1244 (m)	1239 (sh)	1244 (m)	1244 (m)	1251 (sh)	1259 (sh)	1251 (m)		ν (co)
	1199 (w)		1195 (w)		1193 (sh)			1188 (sh)		δ (oh)
1169 (w)		1168 (vw)			1178 (s)	1178 (m)	1179 (w)			δ (ch)
									1050 (w)	
		1143 (w)		1143 (vw)						
	1135 (w)		1131 (w)			1130 (m)	1127 (w)			
					1112 (m)					
1063 (vw)	1082 (vw)	1073 (vw)	1080 (vw)	1075 (vw)		1078 (vw)	1084 (vw)	1082 (vw)		Δ(cc),v (co)
1030 (vw)	1030 (vw)	1030 (vw)	1042 (vw)	1030 (vw)		1030 (vw)	1042 (vw)	1030 (vw)		ν (ring B), γ CH
1002 (vw)	1008 (vw)	1008 (vw)	1008 (vw)	1011 (vw)	1008 (w)	1006 (vw)	1008 (vw)	1008 (vw)		ν (ring B), γ CH
					981 (w)					
									948 (w)	
869 (m)	876 (w)	869 (w)	876 (w)	863 (m)	882 (w)	876 (w)		869 (w)		γ (ch)
									861 (w)	
					838 (m)					
801 (m)	792 (m)	800 (w)	791 (m)			794 (m)	788 (m)	789 (w)		
737 (s)		737 (m)		733 (m)	735 (m)	730 (s)	731 (m)	729 (s)		aromatic system
	715 (s)		715 (s)		707 (w)					
669 (w)	678 (vw)	669 (sh)	677 (w)		673 (w)	679 (m)				
	640 (s)		639 (s)	647 (sh)	636 (sh)	648 (sh)		648 (sh)	648 (w)	Δ(cc)
						630 (s)	632 (s)	633 (s)		Γ (cc)
620 (s)	617 (sh)	620 (s)	614 (sh)	623 (s)	618 (s)					
					573 (w)					
539 (m)	546 (s)	537 (m)	546 (s)	542 (m)	539 (w)	540 (s)	541 (m)	542 (s)		δ (cc)
				492 (sh)	512 (w)	507 (m)				
				479 (w)	471 (m)	476 (m)	478 (w)	479 (m)		Γ (cc)
461 (m)	462 (m)	463 (w)	462 (m)							
428 (sh)		421 (w)		437 (w)	418 (m)			413 (w)		Γ (cc)

Note: ν, stretching; δ, in-plane bending; γ, out-of-plane bending; Γ, skeleton out-of-plane bending; i, inter-ring; Δ, skeleton in-plane bending. S1 (supplier 1); S2 (supplier 2); S3 (supplier 3).

**Table 4 foods-11-03436-t004:** Statistics for each stabilization index in PLSR cross- and internal validations.

	Leave-One-Out Cross Validation			Internal Validation	
	Factors	Corr. Coeff.	RMSECV	Corr. Coeff	RMSEP
Index 1	2	0.91	4.60	0.98	2.16
Index 2	2	0.86	9.26	0.94	6.30
Index 3	2	0.52	0.10	0.88	0.06
Index 4	1	0.17	0.15	0.64	0.11
Index 5	2	0.91	5.94	0.96	4.03
Index 6	2	0.13	0.10	0.79	0.05

## Data Availability

Data is contained within the article or Appendix A.

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
