# Peer review of "Prediction of Anthocyanin Color Stability against Iron Co-Pigmentation by Surface-Enhanced Raman Spectroscopy"

_foods, 2022, doi:10.3390/foods11213436_

Round 1

Reviewer 1 Report

From the viewpoint of statistical analysis, from N=9 samples, why the authors stated in the manuscript:  For the calculation of RMSEP for internal validations, 217 samples were divided into training groups (75%, N=135) and validation groups (25%, N=45) for 218 each index”. The authors should explain in a very detailed way how this data augmentation  process was done. This explanation is a key point to evaluate this paper.  They also should describe the data sets and how these data sets were fed into the algorithms of PCA and PLS analyses. Only with these clear explanations, this paper may be analyzed.

Reviewer 2 Report

Dai et al. have developed a SERS method for prediction of anthocyanin color stability against iron co-pigmentation in 7 different plant sources by using PCA and PLS models. The paper reports an interesting work which has been carried out methodically and should be interest to the food developers. However, my major concern is that the synthesis/characterization of silver colloid and fabrication of SERS substrate are not detailed with sufficient details to aid reproduction by other researchers/scientists in the field. In addition, the following point need to addressed.

1.        In Table 1, the reference numbers should be provided alongside in a separate last column.

2.        In section 2.1, all the chemical and reagents should be provided with the details of state, city and country (in the case of USA) or city and country (in the case of other countries) of company purchase in parenthesis. If the same company is repeating, you no need repeat with these details and just mention the company name.

3.        Line 143, “loaded” will be appropriate word instead of “packed”.

4.        Line 148, “eluded” should be “eluted”.

5.        Section 2.2, if this sample purification method is based on a published article, please cite the reference.

6.        Section 2.5, if the silver nanoparticles preparation method is based on a published article, please cite the reference.

7.        Line 194, “17,000G” should be “17000xg”.

8.        Line 196-197, what is the concentration of silver nanoparticles? Did the authors determine the particle size of silver colloid? Without knowing, this procedure cannot be repeated by others and it seems like silver colloid solution used for diluting the anthocyanin solution to have a concentration of 28 mg/L.

9.        The details of SERS substrate preparation and its characterization is underplayed in this article. I feel that it is the most important information to provide.

10.     L198-199, specify as to how many times the silver colloid-anthocyanin mixture was dropped on aluminum foil coated glass slide for analysis.

11.     How did the authors evaporate the silver colloid-anthocyanin mixture after dropping on glass slide? Please specify the method very clearly without which no one will be able to repeat.

12.     Line 224-226, the details in these lines can also be included as footnote in Table 2.

13.     Line 306-307, provide a reference citation for this statement.

14.     Figure 4, it will be better to mark S1, S2, S3….near the respective clusters inside the PCA scores.

15.     Ensure that all the abbreviations in Tables and Figures are explained in full form in the respective table footnote and figure caption.

16.     Double-check for the uniformity in reference citations, for example Malien-Aubert et al. (2001) in line 381-382 and in other places as [no.].

17.     Line 389, correct the phrase “other the peak”.

Round 2

Reviewer 2 Report

The authors have satisfactorily addressed all the comments raised by reviewers.

Author Response

Thank you for your comments.